# Geostatistical Determination of Ore Shoot Plunge and Structural Control of the Sizhuang World-Class Epizonal Orogenic Gold Deposit, Jiaodong Peninsula, China

**Si-Rui Wang** [1] 🄳**, Li-Qiang Yang** [1,*]**, Jian-Gang Wang** [2]**, En-Jing Wang** [2] **and Yong-Lin Xu** [2]

[1] State Key Laboratory of Geological Processes and Mineral Resources, China University of Geosciences, Beijing 100083, China; creedwang@163.com

[2] Jiaojia Gold Mine, Shandong Gold Mining (Laizhou) Co., Ltd., Laizhou 261400, China; wbin920@sina.com (J.-G.W.); wangej123@163.com (E.-J.W.); xuyonglin@sd-gold.com (Y.-L.X.)

[*] Correspondence: lqyang@cugb.edu.cn; Tel.: +86-(010)-82321937

**Abstract:** The Jiaodong Peninsula in eastern China is the third largest gold-mining area and one of the most important orogenic gold provinces in the world. Ore shoots plunging in specific orientations are a ubiquitous feature of the Jiaodong lode deposits. The Sizhuang gold deposit, located in northwestern Jiaodong, is characterized by orebodies of different occurrences. The orientation of ore shoots has remained unresolved for a long time. In this paper, geostatistical tools were used to determine the plunge and structural control of ore shoots in the Sizhuang deposit. The ellipses determined by variogram modeling reveal the anisotropy of mineralization, plus the shape, size, and orientation of individual ore shoots. The long axes of the anisotropy ellipses trend NE or SEE and plunge 48° NE down the dip. However, individual ore shoots plunge almost perpendicular to the plunge of the ore deposit as a whole. This geometry is interpreted to have resulted from two periods of fluid flow parallel to two sets of striations that we identified on ore-controlling faults. Thrust-related lineations with a sinistral strike-slip component were associated with early-stage mineralization. This was overprinted by dextral and normal movement of the ore-controlling fault that controlled the late-stage mineralization. This kinematic switch caused a change in the upflow direction of ore-forming fluid, which in turn controlled the orientation of the large-scale orebodies and the subvertical plunge of individual ore shoots. Thus, a regional transition from NW-to-SE-trending compression to NW-to-SE-trending extension is interpreted as the geodynamic background of the ore-forming process. This research exemplifies an effective exploration strategy for studying the structural control of the geometry, orientation, and grade distribution of orebodies via the integration of geostatistical tools and structural analysis.

**Keywords:** ore shoots; structural control; geostatistics; Sizhuang gold deposit; Jiaodong Peninsula

## 1. Introduction

Ore shoots are discrete volumes of rock that contain high concentrations of mineralization (particularly high-grade ore). They are commonly hosted within particular structures or related to ore-controlling structures such as faults, shear zones, and folds [1]. The shape, orientation, and distribution of high-grade ore shoots are vital for accurate predictions in the gold mining industry. Ore shoots tend to be elongated in one direction (plunging direction) in lode gold deposits. In most cases, the ore shoot plunge of orogenic gold deposits is the result of mineral deposition and fluid flow direction controlled by structures. The possible structural controls on the localization of ore

shoots include (1) the intersection of a fault or shear zone with a particular lithological unit [1]; (2) the intersection of two syn-metallogenic faults [2]; (3) dilation jogs, divergent bends in faults or zones of en echelon fault segmentation [3]; (4) fold hinge zones [4]; (5) flexures of lode or reef surface, with axes oblique to, and commonly at a high angle to, the movement direction [5]; and (6) zones that plunge subparallel to the striation [6].

More than one control can apply to an individual ore shoot and in which the specific controls on their localization are unclear. The exact plunge and structure control of ore shoots in most mines are hard to ascertain by geological methods alone, because ore shoots show little difference from the surrounding mineralization space in the macro- to mesoscopic view. Some geochemical studies have determined that high-grade ores may be developed with geochemical variations associated with hydrothermal events and enrichment by specific elements, such as Ag, Au, As, Bi, S, Sb, Te, and W. These geochemical indicators and their variations highlight hydrothermal alteration trends and define the plunge of ore shoots [7]. However, these geochemical data cannot access the exact scale and spatial distribution of ore shoots. Nowadays, a combination of structural geology and geostatistics is directly applied in the mining industry. They are integrated to create a method to model structurally controlled deposits numerically. By applying geostatistics, predictions of ore shoots and low-grade zones have been improved and a better understanding of the mechanism of mineralized fluid focus in deposit-scale structures and the results in different ore grades has also been obtained [8]. In this paper, this method is adopted to reveal the plunge of individual ore shoots and the whole deposit.

Sizhuang is a world-class epizonal orogenic gold deposit located in the Jiaodong Peninsula, Northeastern China. It is structurally controlled by the NE-to-NNE-trending and NW-dipping Jiaojia Fault [9]. In Sizhuang, the plunge of orebodies and ore shoots remains unresolved because of the complex architecture of auriferous lodes [10,11].

Thus, lineation measurement, geostatistical analysis, and ore grade data were used to investigate plunge regularity and structural control of high-grade ore shoots in the Sizhaung gold deposit. Finally, structural architecture, grade distribution, and an elaborate model for the formation of ore shoots were summarized.

## 2. Regional and Local Geology

### 2.1. Jiaodong Province and Jiaojia Belt

The Jiaodong gold province is located at the southeastern margin of the North China Craton. It is considered as perhaps the only world-class and giant gold accumulation on Earth, where relatively young gold ores (ca. 126–120 Ma [11–14]) occur within rocks that are at least 2 billion years older (ca. 2.9–1.9 Ga [15,16]) with proven gold resources over 4500 t. Although Jiaodong is commonly considered an orogenic gold province, the mineralization took place later, after regional metamorphism. During an anomalous lithospheric delamination event, the Archean-Paleoproterozoic metamorphic basement of the North China Block was intruded by multiple pulses of Mesozoic granitic magmas [17,18]. The driving force for widespread Late Jurassic and Early Cretaceous granitic magmatism, the switchover from a compressional to an extensional tectonic regime, and gold mineralization are considered to be plate subduction with lithospheric delamination and consequent asthenospheric upwelling [19–21]. Hundreds of gold deposits are widely distributed, but are concentrated in several fault systems with total resources of more than 4500 t gold. Three major NE- to NNE-trending gold belts in western Jiaodong, Sanshandao, Jiaojia, and Zhaoping contain more than 80% of the gold resources of the province and compose the broadly E-to-W-trending world-class gold corridor [22,23] (Figure 1).

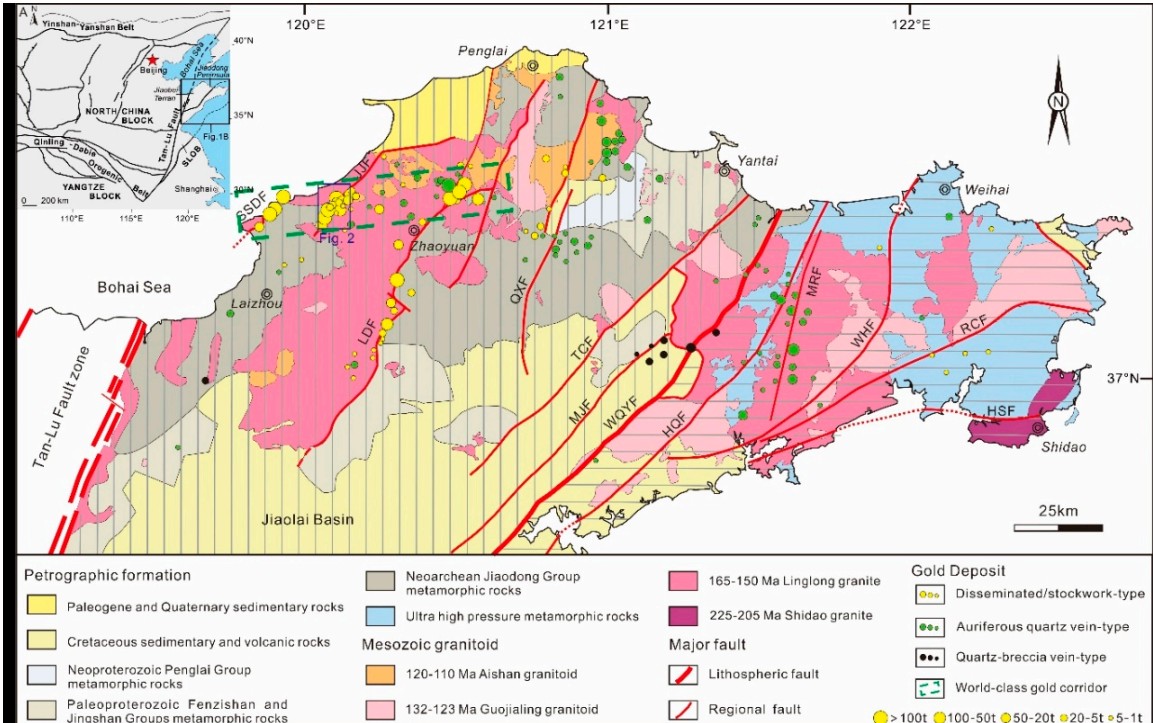

**Figure 1.** Geological map of Jiaodong gold province showing distribution of gold deposits. SSDF, Sanshandao Fault; JJF, Jiaojia Fault; LDF, Linglong Detachment Fault; QXF, Qixia Fault; TCF, Taocun Fault; MJF, Muping-Jimo Fault; WQYF, Wulian-Qingdao-Yantai Fault; HQF, Haiyang-Qingdao Fault; MRF, Muping-Rushan Fault; WHF, Weihai Fault; RCF, Rongcheng Fault; HSF, Haiyang-Shidao Fault [18].

Gold deposits in Jiaodong province have been mainly divided into the extensional massive quartz-vein type and the fracture-fill or disseminated type [24,25]. As the tectonic setting and origin of gold mineralization are different from typical conventional orogenic gold, some studies term these deposits "Jiaodong type" [26–28]. However, the Jiaodong-type gold deposits show no consistent spatial relationship to granitic intrusions of the same age or evidence of metal zonation related to thermal gradients surrounding anomalously hot intrusions in cooler host rocks [29–31]. They do show clear structural control along regional faults [32]. However, the ore and wall rock alteration mineralogy, fluid-inclusion composition, and stable-isotope chemistry are similar to the epizonal orogenic gold deposits [33–36].

The gold deposits in the Jiaojia gold belt, with total resources of >1400 t gold, are mainly controlled by the NE- to NNE-trending and gently NW-dipping Jiaojia and Wang'ershan Faults in an area 50 km from north to south and 1~2 km from east to west [37,38] (Figure 2). These faults underwent different tectonic activities and were normal-sinistral systems during mineralization. Major orebodies of these deposits occur at the footwall of the fault in the SE, characterized by a wide alteration zone and disseminated ores [39–41]. The Jiaojia gold belt is dominated by disseminated or fine-vein ores in the quartz–sericite alteration zone. In addition, gold-bearing quartz–pyrite veins can be found in the potassic alteration zone [42,43].

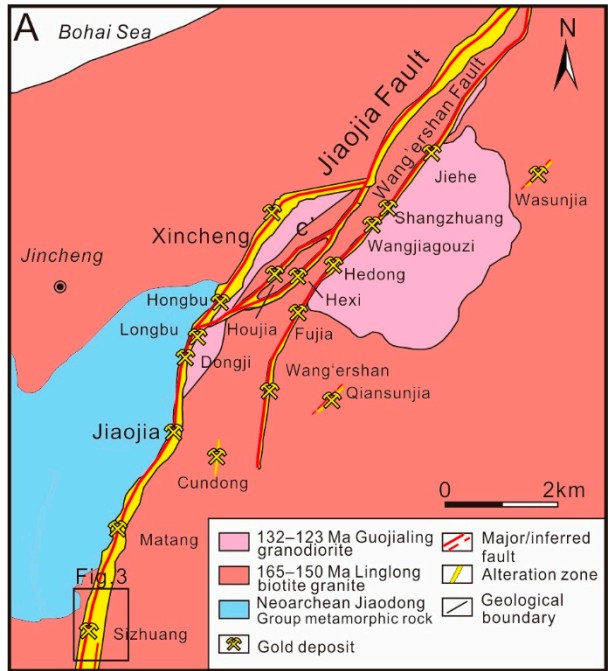

**Figure 2.** Simplified geological map of Jiaojia belt showing distribution of gold deposits [18].

## 2.2. Sizhuang Deposit Geology

The Sizhuang gold deposit (37°20′13″–37°23′28″ N, 120°04′44″–120°08′07″ E) is one of the largest gold deposits in the Jiaodong Peninsula, with a proven resource of >100 t and predicted resource of >200 t. The Sizhaung gold deposit is located about 28 km north of Laizhou City in western Jiaodong and near the southern extremity of the Jiaojia gold field. The Jiaodong group metamorphic rocks (Archean granite–greenstone belt, including TTG (Tonalite-Trondjemite-Granodiorite) gneiss, amphibolites, biotite leptynite, etc.) on the hanging wall of the Jiaojia Fault underwent high-grade metamorphism at about 2.5 Ga. The timing of gold mineralization at the Sizhuang gold deposit is likely to be between 126 and 115 Ma, while the LA-ICPMS (Laser Ablation Inductively Coupled Plasma Mass Spectrometry) zircon U-Pb age 132–123 Ma of the Guojialing granodiorites defines the earliest possible age of hydrothermal activity in the Jiaojia belt. [44,45]. Jurassic Linglong biotite granite is located on the footwall of the Jiaojia Fault to the east of the Sizhuang deposit [46].

The Linglong biotite granite was altered by hydrothermal activity, and altered rock in the footwall of the Jiaojia Fault displays a zoning distribution. Sericite–quartz–pyrite alteration close to the main fault in the footwall has an average width of 20 m. With increasing depth, the altered rocks become thinner. Sericite–quartz alteration is west of the sericite–quartz–pyrite alteration. The outermost part of the alteration zone is a potassic alteration zone, and in the west it is unaltered Linglong biotite granite [27,30,39].

According to their location and geological characteristics, orebodies in the Sizhuang deposit can be divided into three types. No. I orebodies are hosted by sericite–quartz–pyrite altered rock along a fault and are characterized by disseminated ore; No. II orebodies are hosted by sericite–quartz altered rock and are characterized by quartz–sulfide veinlets; and No. III orebodies are hosted by the potassic altered rock and are characterized by quartz–pyrite veins (Figure 3).

The orebodies are discontinuous in the shallow and the deep levels, and can be further divided into two enrichment zones accordingly. The first enrichment zone in the shallow levels is located at a depth from the ground surface to −260 m level, and the second enrichment zone is located at a depth from −310 m to −1000 m. The No. I orebodies strike 20–30°, dip 30–40° to NW overall and are the major components of the first enrichment zone. Ore in the No. I orebodies are characterized by disseminated gold in altered rocks, and tectonic breccias. The No. II orebodies are small, make

up less than 10% of total gold resources in Sizhaung and totally distribute in the first enrichment zone. The second enrichment zone in the deep levels comprises the No. III orebodies and a small part of the No. I orebodies, the No. III orebodies mainly strike NNW-NNE (345°–30°) and dip slightly steeper than the No. I orebodies, with a mean dip angle of 35° [47]. The No. III orebodies are gold bearing pyrite-quartz veins hosted in the secondary en echelon structures in the potassic altered rock. The secondary en echelon fault system was the result of the slip movement of Jiaojia Fault during mineralization [48].

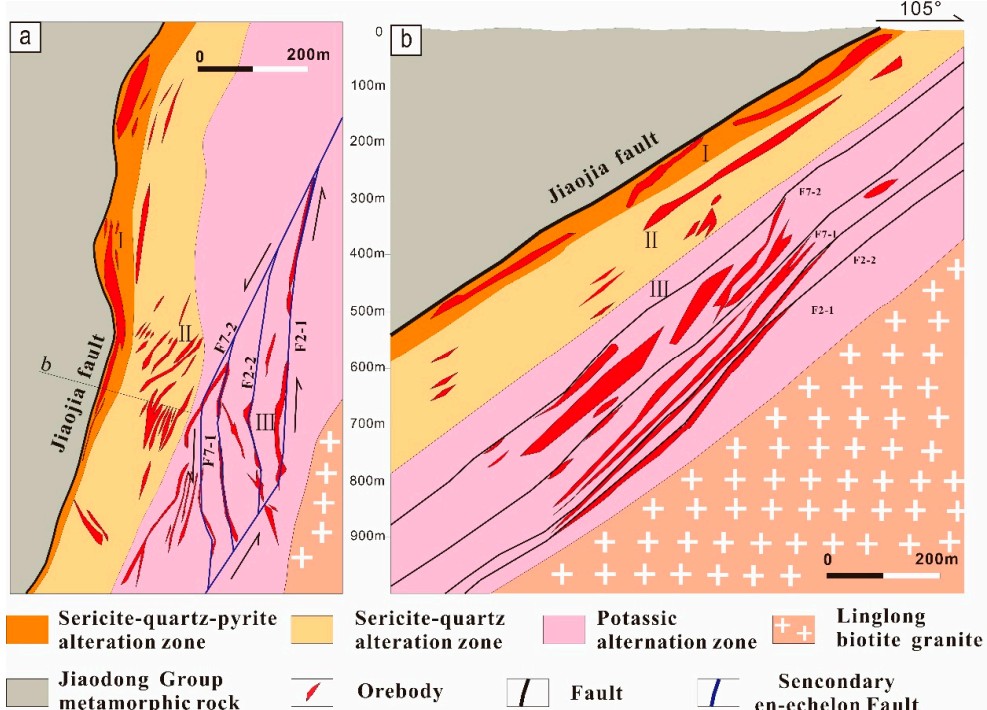

**Figure 3.** (**a**) Geological map of Sizhaung gold deposit, (**b**) NW–SE trending cross-section showing alteration and mineralization distribution. "I", "II" and "III" stand for No. I, No. II and No. III orebodies respectively.

## 3. Gold Grade and Ore-Shoot

### 3.1. Geostatistics Dataset and Analytical Methods

In order to determine the plunge of high-grade ore shoots in the Sizhuang deposit, the geostatistics method and semivariogram function were applied to gold grade distribution analysis.

Ore deposit structural analysis using a combination of structural geology and geostatistics has been directly applied in the mining industry to hydrothermal deposits. The main goal is to integrate structural measurements and assay data to reach the quantitative relation of mineralization and ore-controlling structure. Combined with fault kinematics, metallogenic dynamics can be revealed. Geostatistics is the study of phenomena that vary in space and/or time [49]. It is concerned with the spatial distribution of values, and was originally developed for the purpose of ore deposit modeling and ore reserve estimation [50].

However, the main goal of structural analysis in an ore deposit is to understand the behavior of grade distribution and correlate this information with the structural features of the deposit. The cornerstone of geostatistics is the semivariogram function, which geostatisticians generally refer to as a variogram. Given two locations, x and (x + h), a variogram is one-half the mean square error produced by assigning the value z(x + h) to the value z(x), or the variance of the increment.

The semivariogram function is expressed by the following equation:

$$\gamma(h) = \frac{1}{2N(h)} \sum_{i=i}^{i=N(h)} [z(xi) - z(xi + h)]^2$$

where N(h) is the number of experimental pairs [z(xi), z(xi + h)] of data, separated by the vector h [51].

A total of 8906 samples from bore holes were collected for the study. Most of them were sequential samples with approximately constant volume, taken in regularly spaced lines that cut through sections. Mine levels exist at 45 m intervals. The whole database representing No. I orebodies enclosed a volume of approximately 1000 m to the north, 150 m to the east, and 495 m in depth for −40 m to −535 m levels. The mineralization style of ore in No. I orebodies is disseminated/stockwork, but they have the best continuity as a whole. No. II orebodies are too small and No. III orebodies are too discrete to build up the database, so No. I orebodies were chosen for this study.

The database was multiplied by a constant value to maintain the mining company's confidentiality. This technique does not change the sample variance–range vector relationship and it still allows variographic structural analysis. The aim of variogram structural analysis is to partition the a priori variance and extract the range vectors to determine the anisotropy of the mineralization [52].

Summary statistics and frequency distribution plots were calculated using the moving-window technique to characterize the sample distribution and its general features through the deposit. This procedure indicated that the sample distribution closely followed the lognormal distribution. Therefore, a logarithmic transformation was applied to the dataset to avoid undesirable variations at the sills of the variograms. Frequency distribution plots totally follow normal distribution laws (Figure 4). These initial transformations confine directional variography dictated by the structural elements.

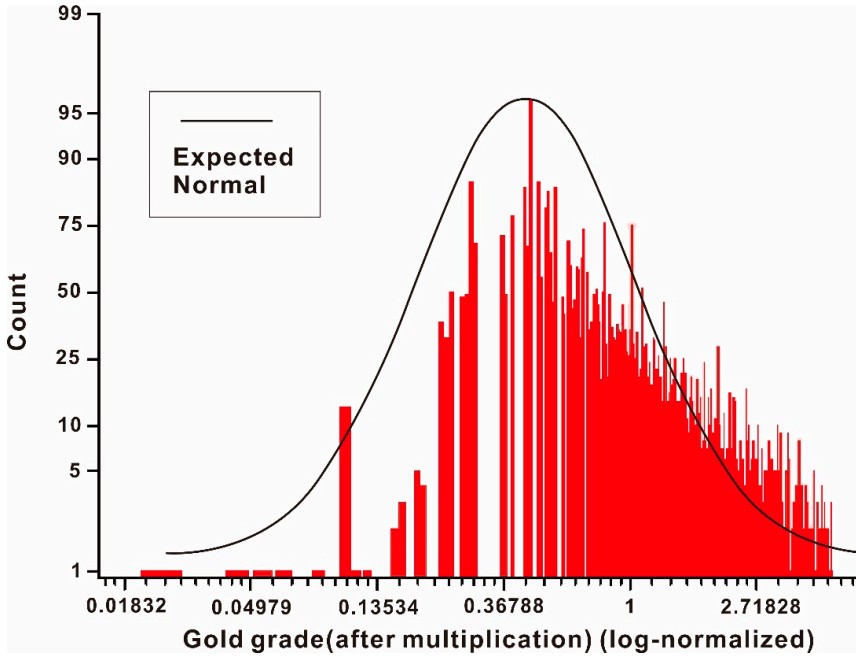

**Figure 4.** Distribution frequency of gold grades for samples from the Sizhuang deposit (logarithm of values). The gold grade presented here had been multiplied and log-normal transformed for confidentiality. The distribution of log-normalized gold grade resembles that of true gold grade, which can be used as the random variable to facilitate the analysis.

### 3.2. Ore-Grade Distribution

Because of the anisotropy of mineralization, gold grades change not only with distance but also direction. The semivariograms obtained by processing the semivariogram function demonstrate the anisotropy [53]. In the interpretation and modeling of variograms, variations in gold grade must be carefully evaluated in different directions, and the backbone of structural variographic analysis is to determine the mechanism by which mineralization spreads in different directions. If the semivariogram continues to climb steadily beyond the global variance value, this indicates a significant spatial trend in the variable. Within the distance of range, the lag distance at which the semivariogram levels off, all grades are autocorrelative. Mineralization anisotropy can be described as an ellipse with a preferred direction of maximum similarity obtained by variograms. The shape of the ellipse is determined by the range calculated from the semivariogram function in different directions. The ellipse that best fits the dataset was determined during variogram modeling. The distance of range shows the distance where gold grade is stable: where the direction range is the biggest, this indicates the best autocorrelation and the gold grade is most stable and changes at the lowest rate. The longer axis of the best-fitting ellipse indicates the orientation and distance where mineralization is stable, and the shorter axis of the ellipse indicates the orientation and distance where mineralization is unstable. In total, the range calculated by the semivariogram function will form an ellipse indicating the variation of gold grade in different directions. This further indicates the orientation and size of high-grade ore shoots as ore shoots plunge in the direction where mineralization is most stable and the range is the biggest [54].

Focusing on the structural geostatistical analysis of grade, general statistics and variograms were carried out. Several tests were conducted to define the parameters to calculate stable planar variograms. Several variographic directions (000°, 020°, 040°, 060°, 080°, 090°, 100°, 120°, 140°, and 160°) were tested. As a result, the directional dependency of the range value (range vector) was obtained during the modeling. The range was used to evaluate the shape of the ellipse that best describes the mineralization anisotropy. An example of variograms calculated for the data from the −40 m level is presented in Figure 5.

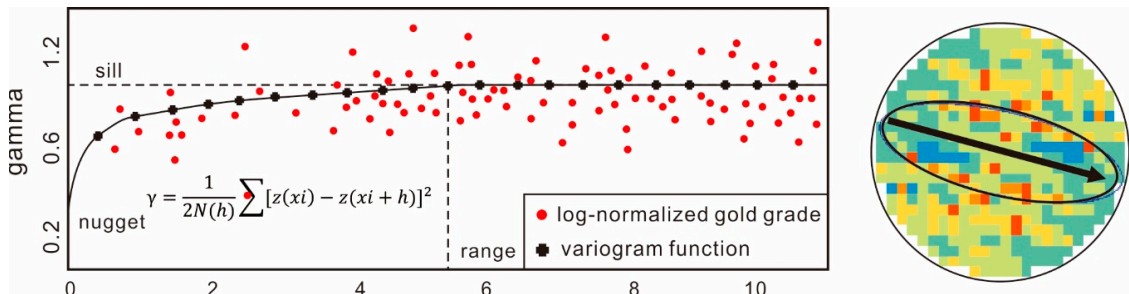

**Figure 5.** Characteristic variogram longitudinal section for −40 m level. The horizontal axis is the sample distance of grade sample. The sill is the value at which the variogram levels off. Range is the lag distance where the semivariogram reaches the sill value and symbolizes the traditional concept of zone of influence of a sample, and represents the distance where the variogram function reaches the sill. The sill is the a priori variance of a random function, or simply the variance of the population. The ellipse on the right describes the mineralization anisotropy with preferred direction.

In addition, different ellipses describing the mineralization anisotropy for the −100 m to −535 m level are presented in Figure 6. In order to clarify the ore shape in three-dimensional space, a variogram in longitudinal section was calculated as well (Figure 7). The parameters determined during variogram modeling are presented in Table 1.

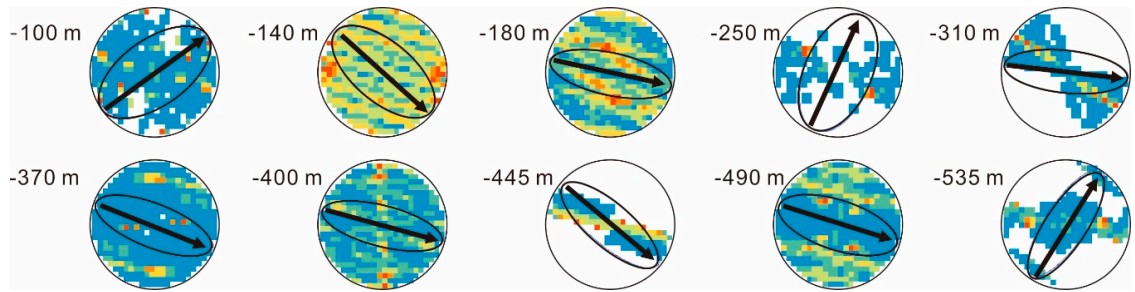

**Figure 6.** Mineralization ellipses to describe ore shape produced by variograms for −100 m to −535 m level.

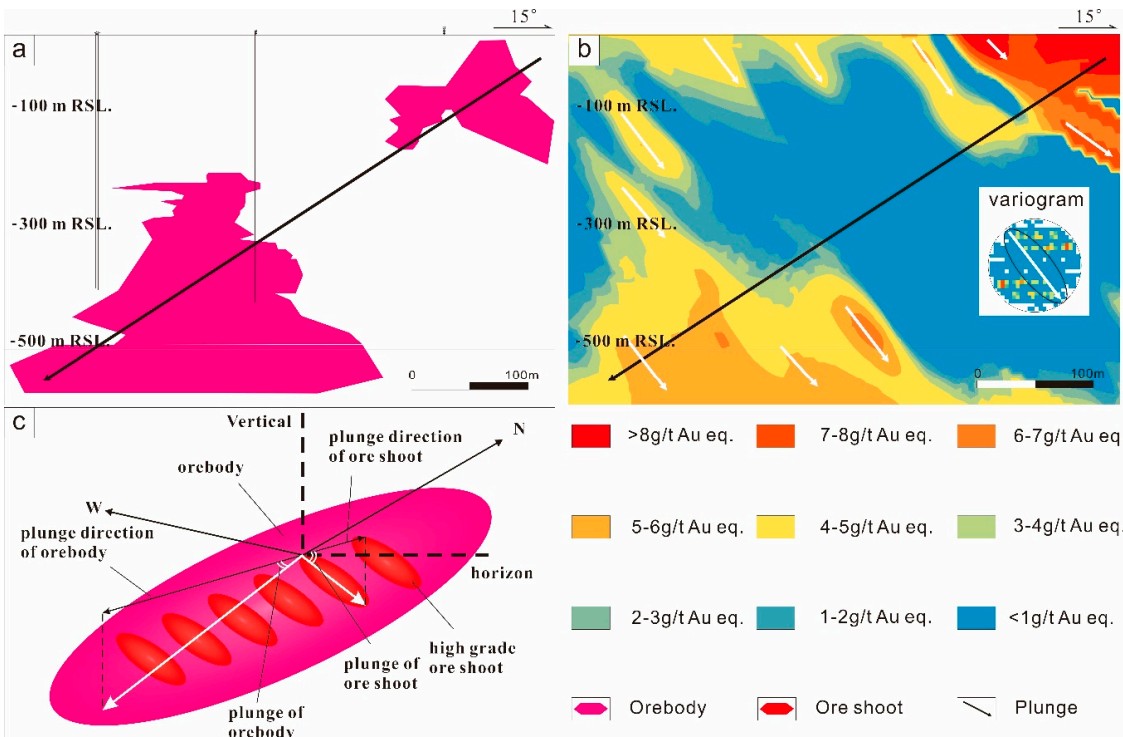

**Figure 7.** Mineralization anisotropy and ore shape in longitudinal section for Sizhuang deposit: (**a**) orebodies in longitudinal section; (**b**) mineralization anisotropy and best-fitting ellipse for ore shape in longitudinal section (black arrow signify plunge of orebodies as a whole, white arrows signify plunge of ore shoots); (**c**) schematic sketch of the geometry of orebodies and ore shoots.

**Table 1.** Parameters determined during variogram modeling.

| Depth (m) | Nugget (g/t) | Maximum Range (m) | Minimum Range (m) | Partial Sill (g/t) | Step (m) | Sill (g/t) | Preferred Direction (°) |
|---|---|---|---|---|---|---|---|
| −40 | 0.59241 | 42.96557 | 14.38454 | 0.243249 | 4.941773 | 0.835659 | 105.6445 |
| −100 | 0.777052 | 2.274927 | 1.241843 | 2.788349 | 0.189577 | 3.565401 | 56.07422 |
| −140 | 2.021469 | 66.53855 | 31.10918 | 0.146952 | 5.544879 | 2.168421 | 129.5508 |
| −180 | 1.111307 | 20.7399 | 6.931124 | 1.340726 | 1.728325 | 2.452033 | 103.3594 |
| −250 | 0.005612 | 8.063422 | 4.352582 | 5.611534 | 0.061969 | 5.617146 | 22.14844 |
| −310 | 0 | 5.251852 | 1.753606 | 95.27344 | 0.437654 | 95.27344 | 95.27344 |
| −370 | 18.34196 | 5.702171 | 1.900767 | 15.11904 | 0.471809 | 33.461 | 112.8516 |
| −400 | 0 | 56.14311 | 18.76465 | 16.14401 | 4.678593 | 16.14401 | 104.0625 |
| −445 | 118.1267 | 12.01075 | 4.027388 | 15.78927 | 1.000896 | 133.916 | 130.0781 |
| −490 | 3.8233 | 10.86118 | 4.2468 | 10.24774 | 0.905099 | 14.07104 | 107.2266 |
| −535 | 7.714254 | 6.450409 | 2.151184 | 7.036681 | 0.537534 | 14.75094 | 33.7422 |
| Longitudinal section | 0 | 87.94258 | 58.94817 | 6.038677 | 7.328548 | 7.328548 | 138.6914 |

### 3.3. Plunge of Orebodies and Fault Kinematics

Orebodies in Sizhuang are generally considered as the products of ore-forming fluids, thus different fluid flow directions would lead to orebodies plunge in different orientations. According to the kinematic indicators of ore-controlling faults planes, the No. I orebodies are identified as the products of early fluid flow and overprinted by late fluid flow, while the No. II and No. III orebodies formed as the products of late fluid flow. In order to constrain the fluid flow that controlled the formation of ore shoots in the No. I orebodies, it was checked whether the plunge of ore shoots in the No. I orebodies matches that of the No. I, No. II and No. III orebodies. Individual orebodies of No. I, No. II, and No. III orebody groups have different orientations, shapes, and sizes (Figure 3). It used to be a problem to ascertain the plunge of individual orebodies. There is a simple and accurate way to determine the maximum, intermediate, and minimum shape axes of an ellipsoidal orebody by extracting and converting the eigenvectors and eigenvalues of the orientations of ellipsoids and put them with the boundaries in three-dimensional space [55]. The plunge and strike of the orebody can then be calculated using simple trigonometric functions and shape axes of ellipsoidal orebodies. The orientation and plunge were plotted in stereoplots (Figure 8). The orebodies in Sizhuang tend to strike to the NNE and plunge to the south on the whole. No. I orebodies strike N-to-NNE and plunge to the SW, and No. II and No. III orebodies strike to the NNW and plunge to the SE.

The ellipses determined by variograms indicate spaces where gold grade distribution is autocorrelative, and these elliptical spaces represent individual ore shoots [5]. The mineralization anisotropy diagrams in Figures 5 and 6 indicate two preferred directions, NE (22–56°) and SEE (95–130°), with an aspect ratio of about 2. This anisotropy reflects the preferential strike of high-grade ore shoots.

However, Figure 7 and Table 1 show that the high-grade ore shoots are general ellipsoid bodies that plunge 48° NE, almost perpendicular to the direction of the orebodies (SW) (Figure 7c). Combining the ellipses determined by variograms in sections with ellipses for each level, a three-dimensional ellipsoid can be determined to describe ore shoot plunge. Ore shoots can be regarded as ellipsoids with almost 100 m maximum axes plunging 48° to the NE, while 60–10 m intermediate axes trend to the NE or SEE and 30–5 m minimum axes vertical to them.

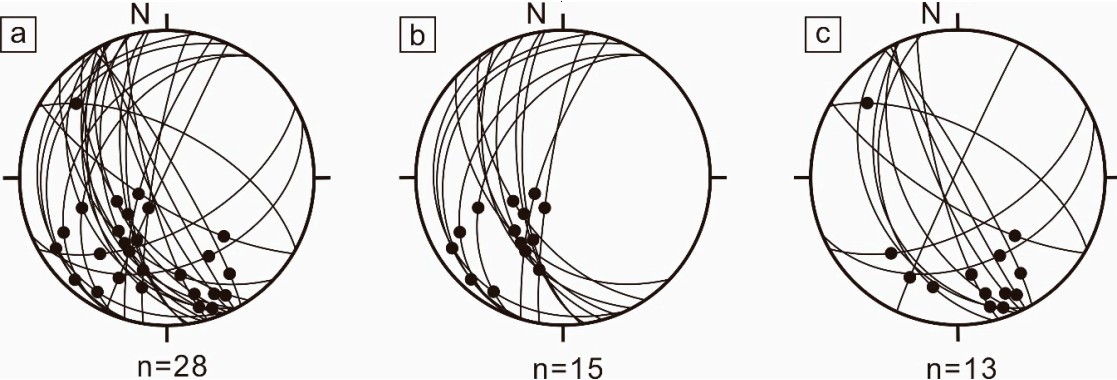

**Figure 8.** (**a**) Stereoplot of all attitudes and plunge calculated from shape axes of all orebodies in Sizhuang deposit. (**b**) Stereoplot of attitudes and plunge of No. I orebodies. (**c**) Stereoplot of attitudes and plunge of No. II and No. III orebodies. All stereoplots are low hemisphere, equal area.

The Sizhuang gold deposit is characterized by a NE- to NNE-trending major fault and secondary en echelon fault system in the footwall (Figure 3). Three concentrations of linear elements were recognized on these fault planes: one at 30°/40° (plunge/trend), one at 20°/190°, and one at 20°/157° (Figure 9c,f). The 30°/40° and 20°/190° planes are fairly close and defined by similar striations on the same rock. They may have been affected by later differential rotations, and we consider them to be one set, and the 20°/157° concentration is different from them. These two sets of linear elements are

defined by striations caused by two different movement activities. The early set is only observed on the major fault and rocks within the sericite–quartz–pyrite alteration zone, which is dominated by No. I orebodies. This set is usually moderately steep and cuts through by later pyrite veins. The late set can be observed within No. I and No. III orebodies. Later lineations consistently trending 150° with shallow dips are usually found on quartz–pyrite veins and almost subparallel to the strike of the ore-controlling fault plane (Figure 9). The two sets of striations indicate early sinistral reverse movement on the ore-controlling faults of the No. I orebodies, followed by dextral normal fault movement on the ore-controlling faults of both the No. I and No. III orebodies.

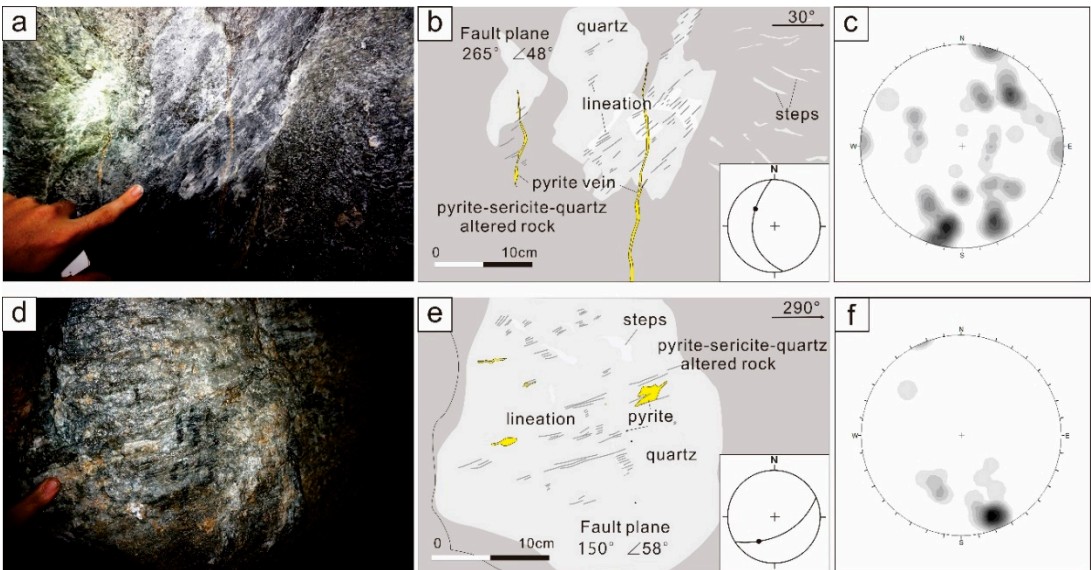

**Figure 9.** (**a**,**b**) Photograph and sketch of slickenfibers related to mineralization in No. I orebodies. (**c**) stereoplot of lineations and slickenfibers indicating movement directions on mineralization controlling fault planes in No. I orebodies. (**d**,**e**) Photograph and sketch of lineations and steps on fault plane. (**f**) Stereoplots of lineations and slickenfibers as movement vector on ore-controlling structure of No. III orebodies. All stereoplots are low hemisphere, equal area.

## 4. Discussion

Although the geostatistics dataset only contains gold grades from No. I orebodies, linear elements indicate that ore shoots of No. I orebodies, No. II and No. III orebodies were controlled by the same fluid flow. The plunge of ore shoots of No. I orebodies imply structural control on No. II and No. III orebodies. Structural control in gold mineralization (Figures 6 and 7) is the result of structurally controlled mineralizing fluid flow [56,57]. At the regional scale of western Jiaodong, the location of gold deposits is commonly controlled by jogs induced by deviations in the regional equivalent stresses. At these places, with resultant heterogeneous strain, rock permeability increased and ore-fluid ingress was focused [58,59]. At the scale of a single orebody, significant gold was deposited in dilational zones in faults. Such fault segments typically result in episodic pressure drop and provide favorable targets for large-tonnage gold ores in giant orogenic gold provinces [60,61].

In Sizhuang, significant gold was deposited in dilational zones in fault zones. Individual ore shoots plunge to the NE while the ore bodies plunge to the SW as a whole. The two main sets of linear elements on faults indicate the upflow of hydrothermal fluids migrating in different directions during two principal mineralization episodes. The directions of fluid flow were same as that of mineralization, and the sequence of episodes caused high-grade mineralization overprinting early wide-ranging mineralization. This led to the formation of SW-plunging orebodies and NE-plunging high-grade ore shoots, in a similar way as has been described for many hydrothermal environments [62,63].

Although hydrothermal flow is complex and poorly constrained, the fault kinematic data suggest a link between high-grade ore shoots and fluid flow controlled by these faults [64]. In addition, the fault movement matches the regional tectonic stress field during mineralization [33,65]. These faults are characterized by multistage events and extensive evolution from compression through transpression to transtension and finally extension, with the transitional phases of transpression and particularly transtension most closely related to gold mineralization.

We propose the following model for ore shoot formation: (1) an early mineralization episode with ore-forming fluids flowing up from SW to NE during sinistral thrusting due to NW-to-SE-trending regional compression that produced SW-plunging orebodies; and (2) a second mineralization episode with ore-forming fluids flowing up from NE to SW along a dextral normal ore-controlling fault as a result of a NW- to SE-trending extensional event that produced NE- plunging orebodies overprinting the early mineralization episode (Figure 10).

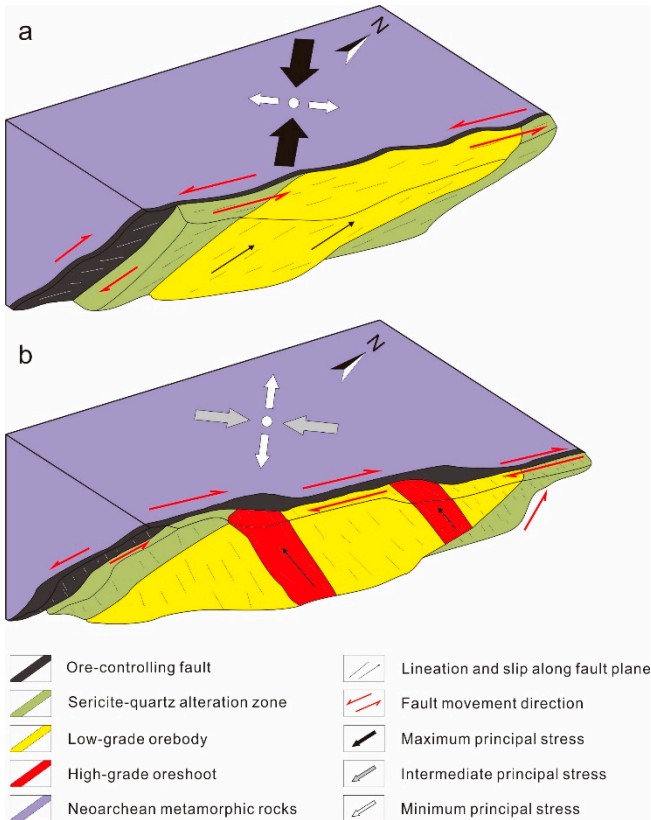

**Figure 10.** Model for structural control of mineralization showing ore-forming fluid flow for Sizhuang gold deposit: (**a**) early mineralization, (**b**) late mineralization.

## 5. Conclusions

The world-class Sizhuang epizonal orogenic gold deposit is strictly fault-controlled. Ellipses of mineralization anisotropy were determined by conducting a spatial analysis of gold grade distribution. Variographic structural analysis shows that high-grade ore shoots tend to plunge to the NE while orebodies plunge to the SW. The geometry and fault kinematics of No. I and No. III orebodies are quite different. No. I orebodies plunging to the SW underwent sinistral thrust and dextral normal movements during the mineralization epoch. No. III orebodies are more likely to plunge to the NE, the same as the plunge direction of high-grade ore shoots. No. III orebodies were controlled by syn-mineralization dextral normal movement. The late ore-forming fluid flow during the late mineralization episode overprinted upon previous alterations and mineralizations. It led to the phenomenon that high-grade ore shoots plunge nearly vertical to the major orebodies.

This study proves that structural analysis combined with geostatistical work is a useful way to determine the architecture of ore-controlling structures at deposit scale. The method could help to design more effective and economic exploration strategies for unknown ore shoots with an emphasis on structural control of the geometry, orientation, and grade distribution of the orebodies.

**Author Contributions:** L.-Q.Y. conceived and designed the ideas; J.-G.W., E.-J.W., and Y.-L.X. provided the geostatistics dataset; S.-R.W. performed the variogram function; L.-Q.Y. and S.-R.W. analyzed the data; S.-R.W. prepared the original draft; S.-R.W. and L.-Q.Y. reviewed and edited the draft.

**Funding:** This study was financially supported by the National Key Research Program of China (Grant No. 2016YFC0600107-4), the National Natural Science Foundation of China (Grant No. 41572069), the MOST Special Fund from the State Key Laboratory of Geological Processes and Mineral Resources, China University of Geosciences (Grant No. MSFGPMR201804), the Key Laboratory of Gold Mineralization Processes and Resource Utilization Subordinated to the Ministry of Natural Resources and Key Laboratory of Metallogenic Geological Process and Resources Utilization in Shandong Province, Shandong Institute of Geological Sciences (Grant No. KFKT201801), and the 111 Project under the Ministry of Education and the State Administration of Foreign Experts Affairs, China (Grant No. B07011).

**Acknowledgments:** We would like to thank Rongxin Zhao and Guangjun Guo at Shangdong Gold Mining for their help during field work, as well as David Groves, Jun Deng, Zhongliang Wang, and Liang Zhang for their help in improving the manuscript.

**Conflicts of Interest:** The authors declare no conflict of interest.

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
