# Peer review of "Geostatistical Determination of Ore Shoot Plunge and Structural Control of the Sizhuang World-Class Epizonal Orogenic Gold Deposit, Jiaodong Peninsula, China"

_minerals, doi:10.3390/min9040214_

Round 1
Reviewer 1 Report
Comments on the manuscript entitled: “Ore Shoot plunge in World-Class Epizonal Orogenic Gold deposit: An example from the Sizhuang Deposit, Jiadong Peninsula, China” submitted to Minerals.
Introduction:
Since the beginning, I have a problem with the use of lineation/striation in the study of falut kinematics. Not so clear, not the same…
Precise also the concept of “ore shoots”….what is precisely? Ok, done in the introduction but you will see further that this remain a problem within the entire manuscript.
Ore shoot always elongated in one direction. Is this the true? Maybe not?
Unclear model of orogenic deposit with respect to IRGD ones….in the introduction.
A lot of question about the figure 1 :
-E-W corridor? Where
-Spatial relations with intrusions….each gold deposit is located within intrusive rocks…Are you sure that gold is not related to these intrusions? Are you sure of the age of the mineralization? Is ther some data? Why this mineralization son't has the age of the 160Ma granite…
The part on the presentation of the Sizhuang deposit, mainly concerned by this study, is unclear and do not contain enough references. At the end, the distribution and characteristics of the orebodies remain unclear in my opinion.
I don’t understand well the concept of variogram calculation in order to obtain the ore grade distribution ellipse? Also, I don’t see the link with speed? But maybe this is because I’m not a specialist but even in this case, the authors have to simplify and to explain clearly, for geologists, the way to process.
Concerning the final model, see my questions about the relationships between fault motions and plunging orebodies/oreshoot? Your model is not enough convincing. Need more explanation, thinking.
In addition, do you mean that in oreshoot, the gold concentration increase with respect to the first orebodies? If so, how do you explain this.
Finally, I thing that this manuscript cannot be published because of the following major points :
- Weak presentation of the geological background
- Bad presentation of the data acquisition and specifically on the determination of elliptical oreshoot by the application of variogram method (and not varoigram….)
- Poorly- convincing model of formation of the two successive gold concentrations with not enough data and argument in order to justify fault motions and formation of dilational jogs.
- As said as a comment in the text, you hve to demonstrate first that the mineralization is syntectonic…and after, you can discuss of the link with oreshoot or orebodies plunging.
In fact, if we are convinced by the structural control (visible rapidly on the cross section of the figure 3b, we are not convinced by the ellipse definition and by the model of formation. There subsists a huge lack of “fact”….kinematic indicators, oreshoot definition, orebodies characterization that remains unclear and badly presented, etc…
Moreover, there subsists a lot of imperfection in the manuscript. I’m not sure that all references are adequately cited? For example, concerning the structural control of ore deposits, some references are lacking, in my opinion. There is also a lot of English mistake and bad syntax.The term "Variogramm" is quasi-never well written.
Please, see also all my comment in the annotated manuscript.

Author Response
Response to Reviewer #1:
Since the beginning, I have a problem with the use of lineation/striation in the study of fault kinematics. Not so clear, not the same…
R: We have improve the quality of photograph used in the article to show stretching lineation more clearly. There are two sets lineation in Sizhuang and their sequence can be indicated by alternation mineral and their relationship. In same domain, the early one showing syn-mineralization dextral normal movement was cut through by the late one showing dextral normal movement.
Precise also the concept of “ore shoots”….what is precisely? Ok, done in the introduction but you will see further that this remain a problem within the entire manuscript.
R: Ore shoots are rock volumes of high grade, the concept differ in type and average grade of mineral deposits. The variogram show relatively high grade ore shoot.
Ore shoot always elongated in one direction. Is this the true? Maybe not?
R: The shape decide whether ore shoot elongated in one direction, oblate bodies (X = Y >> Z) only have strike and dip, not plunge; prolate bodies (X >> Y = Z) have only plunge, not strike nor dip; ellipsoidal bodies (X > Y >Z) can have strike, dip and plunge. In most orogenic gold deposit and this study, ore shoots are ellipsoidal bodies.
Unclear model of orogenic deposit with respect to IRGD ones….in the introduction.
A lot of question about the figure 1 :
-E-W corridor? Where
-Spatial relations with intrusions….each gold deposit is located within intrusive rocks…Are you sure that gold is not related to these intrusions? Are you sure of the age of the mineralization? Is there some data? Why this mineralization son't has the age of the 160Ma granite…
The part on the presentation of the Sizhuang deposit, mainly concerned by this study, is unclear and do not contain enough references. At the end, the distribution and characteristics of the orebodies remain unclear in my opinion.
R: E-W corridor has been added in Figure 1. The intrusions were so close to the mineralization in time and space but previous research indicates ore fluids in Jiaodong are considered most likely to have had a metamorphic source, which is one of the characteristics of orogenic gold deposit. The age of the hosting granodiorites defines the
earliest possible age of hydrothermal activity as they are the host to gold mineralization. The figures and description have been improved as well.
I don’t understand well the concept of variogram calculation in order to obtain the ore grade distribution ellipse? Also, I don’t see the link with speed? But maybe this is because I’m not a specialist but even in this case, the authors have to simplify and to explain clearly, for geologists, the way to process.
R: Variogram calculation is a process to define the mineralization anisotropy. The anisotropy will show in which direction the mineralization are most stable and change at lowest speed. This direction is consider to be the longest axis of ore shoot as well as the plunge direction.
Concerning the final model, see my questions about the relationships between fault motions and plunging orebodies/oreshoot? Your model is not enough convincing. Need more explanation, thinking.
In addition, do you mean that in oreshoot, the gold concentration increase with respect to the first orebodies? If so, how do you explain this.
R: The up-flow of ore-forming fluid is consider to be driven by syn-mineralization fault motions. Different fault movement of early and late mineralization stage lead to the change of fluid migration direction. The late hydrothermal activity overprinting early mineralization made the gold concentration increase and shape the high grade ore shoot which plunge perpendicular to the whole orebody.
Finally, I thing that this manuscript cannot be published because of the following major points :
- Weak presentation of the geological background
- Bad presentation of the data acquisition and specifically on the determination of elliptical oreshoot by the application of variogram method (and not varoigram….)
- Poorly- convincing model of formation of the two successive gold concentrations with not enough data and argument in order to justify fault motions and formation of dilational jogs.
- As said as a comment in the text, you hve to demonstrate first that the mineralization is syntectonic…and after, you can discuss of the link with oreshoot or orebodies plunging.
In fact, if we are convinced by the structural control (visible rapidly on the cross section of the figure 3b, we are not convinced by the ellipse definition and by the model of formation. There subsists a huge lack of “fact”….kinematic indicators, oreshoot definition, orebodies characterization that remains unclear and badly presented, etc…
Moreover, there subsists a lot of imperfection in the manuscript. I’m not sure that all references are adequately cited? For example, concerning the structural control of ore deposits, some references are lacking, in my opinion. There is also a lot of English mistake and bad syntax.The term "Variogramm" is quasi-never well written.
Please, see also all my comment in the annotated manuscript.
R: We appreciate all these comments, we have checked these mistakes and corrected them. We also have modified the presentation of geology background, added schematic sketch the geometry of orebodies and ore shoots in Figure 7. More modification can be seen in the Word document attached.

Reviewer 2 Report
An interesting study is presented combining geostatistical analysis of essay data (variograms) to determine the geometry of minerallized zones in the Sizhuang Au ore deposit, with analysis of fault planes striations and slickenfibers. The 2 sets of data are integrated in an attractive genetic model that includes two episodes of minerallization. The conclusions are reasonably well supported by the presented data.
The paper requires further attention before it can be published. I have edited parts of it, but more need to be done along the same lines. This concerns the form not the contents, so I recommend publication after a moderate revision. I attach a separate Word document with detailed comments and criticisms.

Author Response
Response to Reviewer #2:
An interesting study is presented combining geostatistical analysis of essay data (variograms) to determine the geometry of minerallized zones in the Sizhuang Au ore deposit, with analysis of fault planes striations and slickenfibers. The 2 sets of data are integrated in an attractive genetic model that includes two episodes of minerallization. The conclusions are reasonably well supported by the presented data.
The paper requires further attention before it can be published. I have edited parts of it, but more need to be done along the same lines. This concerns the form not the contents, so I recommend publication after a moderate revision. I attach a separate Word document with detailed comments and criticisms.
R: Thanks for your comments, we have modified the form and the errors your criticisms mentioned. We have added the reference and improve Figure 1,3,4,7 as well as some description about geology background and variogram method. Detail modification can be seen in the Word document attached.

Round 2
Reviewer 1 Report
In my opinion, the consideration of my remarks of the first review is not sufficiently taken into account. I underline some part in the joint document for which no change has been made.
I appreciate the addition of the figure 7c but I think that a lot a problem are not solved or are ignored ny the authors such as, for example, the lineation/striation problem, the chronological relationships and some other points in yellow in the annotated text.
A little effort in considering all these points can help for the publication of this paper. For that, I recommend a minor revision but a real revision.

Author Response
Thanks for your comments. The lineation on rocks are mostly striation cause by fault movement. We change the word 'stretching lineation' to 'striation' in the manuscript. The en-echelon fault system can be seem in plane view at the footwall of Jiaojia fault and we modified secondary en-echelon fault system in Figure 3 to make it clear. Some description has been improved as well.
